# Temporal Trends in Mortality Associated with Comorbid Type 2 Diabetes and Schizophrenia: The Fremantle Diabetes Study

**DOI:** 10.3390/jcm12010300

**Published:** 2022-12-30

**Authors:** Wendy A. Davis, David G. Bruce, Sergio E. Starkstein, Timothy M. E. Davis

**Affiliations:** Medical School, University of Western Australia, Fremantle Hospital, P.O. Box 480, Fremantle, WA 6959, Australia

**Keywords:** type 2 diabetes, schizophrenia, mortality

## Abstract

Background: In Phase I of the community-based Fremantle Diabetes Study (FDS1), there was evidence of a deleterious interactive effect of schizophrenia and type 2 diabetes on mortality. Our aim was to investigate whether the mortality gap had improved in FDS Phase II (FDS2) conducted 15 years later. Methods: Participants with type 2 diabetes from FDS1 (n = 1291 recruited 1993–1996) and FDS2 (n = 1509 recruited 2008–2011) were age-, sex- and postcode-matched 1:4 to people without diabetes. Schizophrenia at entry and incident deaths were ascertained from validated administrative data. Results: Schizophrenia affected 50/11,195 (0.45%) of participants without diabetes and 17/2800 (0.61%) of those with type 2 diabetes (*p* = 0.284). During 142,304 person-years of follow-up, the mortality rate (95% CI) was lowest for the FDS2 subgroup without diabetes/schizophrenia (18.2 (16.9, 19.6)/1000 person-years) and highest in FDS2 and FDS1 subgroups with type 2 diabetes/schizophrenia (53.3 (14.5, 136.6) and 98.0 (31.8, 228.8)/1000 person-years, respectively). Compared to the respective FDS subgroup without diabetes/schizophrenia, the mortality rate ratio was approximately 50% higher in the type 2 diabetes subgroup, and three times higher in those with type 2 diabetes/schizophrenia. In Cox regression, unadjusted hazard ratios were highest in those with type 2 diabetes/schizophrenia in FDS1 (HR (95% CI): 3.71 (1.54, 8.93) and FDS2 (2.96 (1.11, 7.91)), increasing to 5.61 (2.33, 13.5) and 26.9 (9.94, 72.6), respectively, after adjustment for age. Conclusions: Although limited by small numbers of schizophrenia cases, these data suggest that comorbid type 2 diabetes and schizophrenia remains associated with a substantial and possibly increasing mortality gap.

## 1. Introduction

There are strong associations between schizophrenia and type 2 diabetes [1]. People with schizophrenia are especially prone to developing type 2 diabetes through a combination of suboptimal lifestyle factors, metabolic and obesogenic effects from antipsychotic medications, and a possible shared genetic predisposition [2,3,4]. Schizophrenia sufferers have a substantially increased risk of death compared with the general population. This gap reflects medical rather than psychiatric illness [5,6], including a heightened risk of cardiovascular disease (CVD) explained by relatively adverse vascular risk factors and high rates of cigarette smoking [7,8]. Although relevant studies have had important methodological differences, the risk of death in people with comorbid type 2 diabetes and schizophrenia appears especially large [9,10,11]. These poor outcomes may be due to suboptimal schizophrenia management coupled with under-diagnosis and under-treatment of coincident type 2 diabetes [12].

In a previous longitudinal observational study, the Fremantle Diabetes Study Phase I (FDS1), which began in the early 1990s, we explored the mortality gap associated with schizophrenia and type 2 diabetes [9]. Comorbid schizophrenia and type 2 diabetes increased the risk of dying more than three-fold compared with type 2 diabetes alone and more than six-fold compared with people without either disorder, consistent with other population-based studies [13]. The aim of the present study was to investigate whether the mortality gap had improved when a similar study in the same population, the FDS Phase II (FDS2), was conducted 15 years later. While this interval may appear short, it spans a time of considerable change in clinical psychiatric practice including increased recognition of the importance of physical health measures in the management of chronic mental disorders [14,15] but increased use of second-generation antipsychotic agents and their associated adverse metabolic effects [16] in countries such as Australia [17,18], set against improvements in cardiovascular risk in type 2 diabetes in FDS2 compared with FDS1 [19].

## 2. Materials and Methods

### 2.1. Study Participants

FDS1 and FDS2 were similarly conducted, observational, longitudinal studies of known diabetes conducted in the same zip code-defined geographic area surrounding the port city of Fremantle in Western Australia (WA) [20,21]. FDS1 recruited participants between 1993 and 1996 and FDS2 between 2008 and 2011. Both studies utilized the same strategy to identify all potential participants with known diabetes in the area from hospital, clinic and primary care patient lists, widespread advertising through local media, pharmacies, optometrists, networks of health care professionals, but, in the case of FDS2, third-party mail-outs to registrants of the Australian National Diabetes Services Scheme and the National Diabetes Register were also employed [21]. Details of recruitment, sample characteristics including classification of diabetes types, and non-recruited people identified with diabetes in the catchment area have been published [20,21]. The FDS1 protocol was approved by the Fremantle Hospital Human Rights Committee, and the FDS2 protocol by the Human Research Ethics Committee of the Southern Metropolitan Area Health Service. All participants gave written informed consent. The WA Data Linkage System (WADLS) [22] was used to match FDS1 and FDS2 participants with residents in the study catchment area who did not have documented diabetes and to obtain further clinical data. Linkages with clinical databases were approved by the WA Department of Health Human Research Ethics Committee.

In FDS1, 2258 people with diabetes were identified from a population of approximately 120,000, and 1426 (63%) recruited of whom 1296 (91%) had clinically defined type 2 diabetes. In FDS2, 4639 people with diabetes were identified from a population of approximately 157,000, and 1668 (36%) recruited of whom 1509 (90%) had type 2 diabetes. Four age-, sex- and postcode-matched residents without any documentation of diabetes were randomly selected from the study catchment area for each FDS1 and FDS2 participant at the time of their recruitment using the Western Australian Electoral Roll as a source of all adults resident in the FDS catchment area and, for FDS2, the WA Registry for Births, Deaths and Marriages. In FDS1, five residents died just before their matched participant with diabetes was enrolled and were therefore excluded. In addition, matches could not be made for five young and four elderly FDS1 participants who were also excluded, leaving 1291 participants with type 2 diabetes (99.6%) matched with 5159 residents without diabetes. In FDS2, the 1509 FDS2 participants with type 2 diabetes were matched with 6036 residents without diabetes.

### 2.2. Baseline Assessment

For those with type 2 diabetes, the assessment at study entry in FDS1 and FDS2 included comprehensive questionnaires, a physical examination and relevant tests including electrocardiography, and fasting biochemical tests performed in a single nationally accredited laboratory (18). In the case of participants without diabetes, there were no face-to-face assessments and data were restricted to hospitalizations, use of mental health services, and deaths available through the WA Data Linkage System (WADLS).

### 2.3. Ascertainment of Schizophrenia, Comorbidities and All-Cause Mortality

Linkage through the WADLS provided validated data on prior history and incident events. In WA, the Hospital Morbidity Data Collection (HMDC) contains information regarding all public/private hospitalizations since 1970, the Mental Health Information System (MHIS) started as a register of psychiatric inpatients in 1966, was expanded to include hospitals and community mental health services in the 1970s and, since 1980, has covered all outpatient, community-based and hospital-based mental health services, and the Death Register contains information on all deaths [22]. Participants were followed to death or end-2012 in FDS1 and end-2016 in FDS2, whichever came first. In addition, participants in the matched cohort without diabetes at study entry were censored if they subsequently developed diabetes at the date of the first record of diabetes in any of the linked databases. Participants in both cohorts without schizophrenia at study entry were censored if they subsequently developed schizophrenia at the date of the first record of schizophrenia in any of the linked databases. Relevant International Classification of Disease (ICD)-9-CM (295) and ICD-10-AM diagnosis codes (F20) were used to identify prevalent and incident schizophrenia in the HMDC and MHIS. The HMDC was also used to supplement data obtained through FDS assessments relating to prevalent/prior disease. Using the HMDC, the Charlson Comorbidity Index (CCI) was calculated for the five years prior to study entry [23], excluding codes specific to diabetes and its complications. This early version of the CCI did not include mental health conditions.

### 2.4. Statistical Analysis

The computer packages IBM SPSS Statistics 25 (IBM Corporation, Armonk, NY, USA) and StataSE 15 (College Station, TX: StataCorp LP) were used for statistical analysis. Data are presented as proportions or mean ± standard deviation (SD). Two-sample comparisons were by Fisher’s exact test for proportions and Student’s *t*-test for normally distributed variables. The Bonferroni correction was used to adjust for multiple comparisons.

For each FDS phase, mortality rates were calculated for each of four subgroups categorised by type 2 diabetes and schizophrenia status (type 2 diabetes alone, schizophrenia alone, both conditions, neither condition). The neither condition subgroup, which had the lowest mortality rate, was used as the comparator for the calculation of mortality rate ratios and mortality rate differences for each of FDS1 and FDS2. Cox proportional hazards modelling was performed to determine how the survival of people with type 2 diabetes and without diabetes by schizophrenia status differed in each study phase. Models were performed unadjusted and adjusted for age, and additionally for sex and CCI. The proportional hazards assumption was checked using time-varying covariates. When this assumption was violated, a time-varying interaction of each covariate with ln(time) was added to the model.

## 3. Results

### 3.1. Participant Characteristics

Schizophrenia was uncommon, affecting 67/13,995 (0.48%) people overall; 17/2800 (0.61%) of those with type 2 diabetes and 50/11,195 (0.45%) without diabetes (*p* = 0.284). The baseline characteristics of participants in the two FDS phases categorized by diabetes and schizophrenia status are summarized in Table 1. Compared with all FDS1 participants (i.e., those with type 2 diabetes plus the matched comparison group), the FDS2 participants were slightly but significantly older (65.4 ± 11.7 versus 64.0 ± 11.2 years, *p* < 0.001). In both FDS1 and FDS2, participants with schizophrenia were significantly younger than those without (FDS1: 57.4 ± 11.0 versus 64.0 ± 11.2 years, *p* = 0.003; FDS2: 58.1 ± 13.6 versus 65.5 ± 11.6 years, *p* = 0.001). Participants with comorbid schizophrenia and type 2 diabetes were considerably younger than those with type 2 diabetes alone (47.8 ± 10.9 versus 65.6 ± 11.6 years, *p* < 0.001 Bonferroni-corrected) in FDS2, but this was not the case in FDS1 (61.7 ± 8.0 versus 64.0 ± 11.2 years, *p* > 0.999 Bonferroni-corrected). In FDS1 and FDS2, participants with type 2 diabetes had significantly more comorbidities than those with neither condition (both *p* < 0.001 Bonferroni-corrected). There was no statistically significant excess of comorbidities in participants with schizophrenia, whether or not they had type 2 diabetes (*p* ≥ 0.183 Bonferroni-corrected).

### 3.2. Mortality by Type 2 Diabetes and Schizophrenia Status, and Study Phase

In FDS1, participants with type 2 diabetes and without diabetes were followed until death or end-2012 for 16,723 person-years (mean ± SD 13.0 ± 6.1 years) and 75,541 person-years (mean ± SD 14.6 ± 5.7 years), respectively. In FDS2, participants with type 2 diabetes and without diabetes were followed until death or end-2016 for 10,070 person-years (mean ± SD 6.7 ± 1.7 years) and 39,972 person-years (mean ± SD 6.6 ± 1.8 years), respectively. Table 1 provides comparative mortality data (deaths, mortality rates, mortality rate differences, mortality rate ratios, hazard ratios) for the four sub-groups of participants (no diabetes/no schizophrenia, type 2 diabetes/no schizophrenia, no diabetes/schizophrenia, type 2 diabetes/schizophrenia) in each study (FDS1 and FDS2). The mortality rate (95% CI) was lowest for the FDS2 subgroup without diabetes/schizophrenia (18.2 (16.9, 19.6)/1000 person-years) and highest in both the FDS2 and FDS1 subgroups with type 2 diabetes/schizophrenia (53.3 (14.5, 136.6) and 98.0 (31.8, 228.8)/1000 person-years, respectively). Using the respective FDS subgroup without diabetes/schizophrenia as the comparator, the mortality rate ratio was approximately 50% higher in the type 2 diabetes subgroup in both phases, and three times higher in those with comorbid schizophrenia and type 2 diabetes (see Table 1).

Cumulative mortality plots for the four sub-groups for each Phase considered separately are shown in Figure 1.

In both FDS1 and FDS2, the presence of schizophrenia increased the risk of death in those with type 2 diabetes, especially after the first 6 years of follow-up, but this was statically significant only for FDS1 (*p* = 0.045 by log rank test; estimated median survival (95% CI): 16.2 (15.2, 17.2) vs. 8.0 (7.5, 8.4) years in those without versus with schizophrenia).

In Cox proportional hazards modelling utilizing the same comparator groups, the unadjusted hazard ratios were highest in those with comorbid schizophrenia and type 2 diabetes in both FDS1 (HR (95% CI): 3.71 (1.54, 8.93) and FDS2 (2.96 (1.11, 7.91)), increasing to 5.61 (2.33, 13.5) and 26.9 (9.94, 72.6), respectively, after adjustment for age. Further adjustment for sex and the CCI and then the time-varying interactions of these covariates with ln(time) had a modest effect (Table 1).

## 4. Discussion

The present study provides evidence of substantial temporal changes in mortality in the 15 years between FDS phases in some groups defined by type 2 diabetes and schizophrenia status. There were improvements in absolute mortality rates in all subgroups except for schizophrenia without type 2 diabetes. However, in Cox models, the unadjusted risk of death continued to be approximately 50% higher in the type 2 diabetes without schizophrenia subgroup and three-fold higher in the type 2 diabetes with schizophrenia subgroup compared with the subgroup without type 2 diabetes or schizophrenia. After adjustment for age, the risk of death in those with type 2 diabetes and schizophrenia increased to a six-fold greater risk in FDS1 and a 27-fold greater risk in FDS2 compared with those with neither condition. Despite the wide confidence intervals, we conclude that the combination of type 2 diabetes and schizophrenia remains associated with a substantial and possibly increasing mortality gap.

The improvements in longevity seen in those without schizophrenia are consistent with the experience reported elsewhere with national samples [24,25,26] and with our own finding of improvements in cardiovascular disease incidence and mortality between the two phases of the FDS [19]. The lack of improvement in the mortality gap seen with schizophrenia alone and the possibly increasing gap with comorbid type 2 diabetes is consistent with recent published studies [27,28]. There is evidence that people with schizophrenia have improving life expectancy in high-income countries although insufficient to reduce the mortality gap compared with the general population [27]. Consequently, the apparent worsening of the mortality risk in those with comorbid disease in FDS2 may reflect a relative lack of improvement rather than a worsening health status. Nevertheless, the combination of type 2 diabetes and schizophrenia remains associated with a substantially shortened life span and the current study finds little evidence of observable improvements in the health of this particularly vulnerable patient group.

A detailed assessment of the reasons why people with comorbid schizophrenia and type 2 diabetes in the present study do not appear to have benefited from improved contemporary management was beyond the scope of the present study. It would be of relevance to examine whether there were between-group temporal differences in the proportions achieving cardiometabolic targets (glycated haemoglobin, blood pressure and serum lipid levels) and developing obesity, including their potentially important relationship with antipsychotic medication use in those with schizophrenia, whether smoking rates had remained relatively high in those with schizophrenia, and whether health service utilization including screening for complications was influenced by both type 2 diabetes and schizophrenia status. There are some relevant data available through the comprehensive screening of FDS participants but the confidential matching of people without diabetes in the study catchment area means that only simple administrative data relating to endpoints are available for these participants.

The present study had other limitations. There was an understandably low number of schizophrenia cases overall which is reflected in the wide risk estimate confidence intervals. Nevertheless, the prevalence in our participants without diabetes (0.45%) was in the range (0.28% to 0.45%) reported in other general population studies [29,30]. In addition, we identified those with schizophrenia, and confirmed the diagnosis, using the MHIS, a database validated previously by comparison with semi-structured interviews which showed that a registered diagnosis of schizophrenia had a sensitivity of 0.92 and specificity of 0.88 [31]. There may be more unrecognized type 2 diabetes complicating schizophrenia than undiagnosed type 2 diabetes in the general population [32] which would help explain the similar baseline prevalence rates in participants with type 2 diabetes and the combined sample as a whole. Both phases of the FDS required participants to volunteer their involvement and attend repeat assessments, raising the possibility of recruitment and survivor bias. It is possible that the two FDS phases recruited milder schizophrenia cases with comorbid type 2 diabetes, suggesting that the mortality gap exhibited in the present study may have been even greater had more severe cases been captured. There were different periods of follow-up in the two study phases which may have influenced comparisons but the patterns of relative mortality shown in Figure 1 appeared consistent across FDS1 and FDS2. The study strengths include the representative nature of the FDS in terms of Australian urban diabetes, the comprehensive nature of the assessments, the length of follow-up of the study participants and, importantly, the similar nature of the recruitment and follow-up methods used during the two phases of the study.

In conclusion, FDS data collected over a long duration of follow-up during two distinct time periods 15 years apart allowed the assessment of temporal trends in all-cause mortality in a large community-based sample by type 2 diabetes and schizophrenia status. Overall improvements in mortality were seen in FDS2 compared with FDS1, but the mortality gap due to schizophrenia was evident in both studies and the combination of type 2 diabetes and schizophrenia was associated with a substantially increased risk of death. This latter observation is likely multifactorial, perhaps reflecting less effective cardiometabolic management in those with schizophrenia compounded by the adverse effects of antipsychotic therapies. The relatively young age of those with schizophrenia and type 2 diabetes in FDS2 compared with other subgroups suggests that the recent increased use of second-generation antipsychotic agents with obesogenic and associated adverse metabolic effects has contributed to an earlier emergence of comorbid disease. Regardless of possible causes, patients with both type 2 diabetes and schizophrenia comprise a vulnerable group with a particularly poor prognosis. Further efforts, including through additional larger-scale population-based studies involving robust methods of ascertainment of both diagnoses, are urgently required to confirm the present findings and characterize and thus help address the unmet health needs of individuals with comorbid schizophrenia and type 2 diabetes.

## Figures and Tables

**Figure 1 jcm-12-00300-f001:**
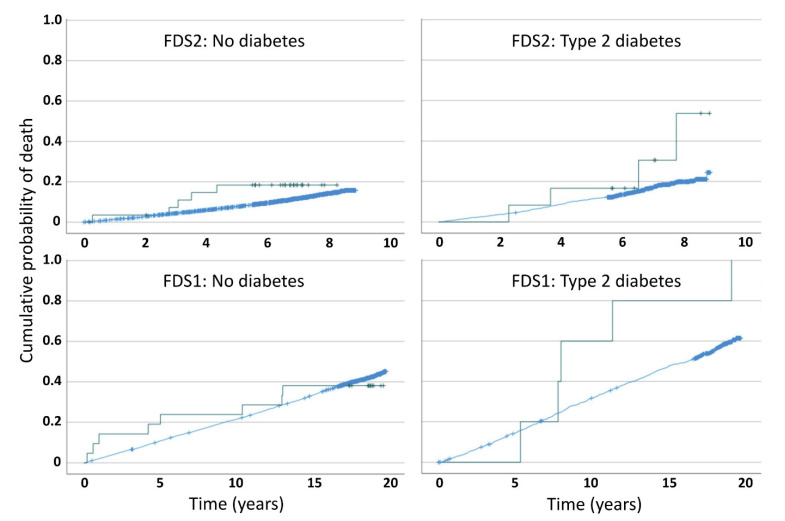
Cumulative mortality estimates by type 2 diabetes and schizophrenia status for each Fremantle Diabetes Study Phase. In each panel, no schizophrenia is the blue line and schizophrenia the green line. Censored points are shown as +.

**Table 1 jcm-12-00300-t001:** Baseline characteristics and subsequent mortality in participants with type 2 diabetes and matched individuals without diabetes by prevalent schizophrenia status for each FDS Phase.

		FDS1				FDS2		
	No Diabetes, No Schizophrenia	T2D, No Schizophrenia	Schizophrenia, No Diabetes	T2D, Schizophrenia	No Diabetes, No Schizophrenia	T2D, No Schizophrenia	Schizophrenia, No Diabetes	T2D, Schizophrenia
Number (% by Phase)	5138 (79.7)	1286 (19.9)	21 (0.3)	5 (0.1)	6007 (79.6)	1497 (19.8)	29 (0.4)	12 (0.2)
Age at FDS entry (years)	64.0±11.2	64.0±11.2	56.4±11.6	61.7±8.0	65.4±11.7	65.6±11.6	62.4±12.3	47.8±10.9
Sex (% male)	48.8	48.7	38.1	60.0	51.8	51.8	48.3	58.3
CCI (%):								
0	85.5	71.8	90.5	40.0	86.5	75.0	89.7	91.7
1 or 2	11.2	21.9	9.5	60.0	9.8	17.0	10.3	0
≥3	3.3	6.4	0	0	3.7	8.0	0	8.3
Follow-up (person-years)	75,253	16,672	287	51	39,807	9,995	164	75
Deaths (n (%))	2151 (41.9)	729 (56.7)	8 (38.1)	5 (100)	725 (12.1)	265 (17.7)	5 (17.2)	4 (33.3)
Mortality rate (/1000 person years)	28.6 (27.4, 29.8)	43.7 (40.6, 47.0)	27.9 (12.0, 54.9)	98.0 (31.8, 228.8)	18.2 (16.9, 19.6)	26.5 (23.4, 29.9)	30.5 (9.9, 71.2)	53.3 (14.5, 136.6)
Mortality rate difference (/1000 person years)	-	15.1 (11.8, 18.5)	−0.71 (−20.1, 18.6)	69.5 (−16.5, 155)	-	8.30 (4.84, 11.8)	12.3 (−14.5, 39.0)	35.1 (−17.2, 87.4)
Mortality rate ratio (95% CI) (reference no diabetes, no schizophrenia)	1.00	1.53 (1.40, 1.66)	0.98 (0.42, 1.92)	3.43 (1.11, 8.02)	1.00	1.46 (1.26, 1.68)	1.67 (0.54, 3.92)	2.93 (0.80, 7.53)
HR (95% CI) unadjusted	1.00	1.56 (1.44, 1.70)	0.97 (0.48, 1.94)	3.71 (1.54, 8.93)	1.00	1.45 (1.26, 1.67)	1.74 (0.72, 4.18)	2.96 (1.11, 7.91)
HR (95% CI) adjusted for age	1.00	1.71 (1.57, 1.86)	2.27 (1.13, 4.54)	5.61 (2.33, 13.5)	1.00	1.46 (1.27, 1.69)	2.30 (0.95, 5.54)	26.9 (9.94, 72.6)
HR (95% CI) adjusted for age, sex and CCI	1.00	1.55 (1.42, 1.69)	2.87 (1.43, 5.75)	4.77 (1.98, 11.5)	1.00	1.23 (1.07, 1.42)	2.85 (1.18, 6.89)	25.6 (9.46, 69.3)
HR (95% CI) adjusted for age, sex, CCI and their time-varying interactions with ln(time)	1.00	1.54 (1.41, 1.68)	2.82 (1.40, 5.65)	4.52 (1.88, 10.9)	1.00	1.24 (1.07, 1.43)	2.90 (1.20, 6.99)	24.7 (9.09, 67.0)

FDS1 Fremantle Diabetes Study Phase I; FDS2 Fremantle Diabetes Study Phase II; T2D type 2 diabetes; CCI Charlson Comorbidity Index calculated for the 5 years prior to study entry, excluding codes specific to diabetes and its complications; HR hazard ratio.

## Data Availability

The exposure, outcome and matched data that support the findings of this study are available from the Western Australian Department of Health but restrictions apply to the availability of these data, which were used under strict conditions of confidentiality for the current study, and so are not publicly available. Data are however available from the authors upon reasonable request and with permission of Western Australian Department of Health.

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
