# Peer review of "Temporal Trends in Mortality Associated with Comorbid Type 2 Diabetes and Schizophrenia: The Fremantle Diabetes Study"

_jcm, 2022, doi:10.3390/jcm12010300_

Round 1
Reviewer 1 Report
This study aims at evaluating the temporal changes in mortality in different groups defined by type 2 diabetes and schizophrenia. The topic is very relevant and manuscript is overall well written. A strength is the large dataset with patients recruited in different time periods including sufficient numbers of diabetes patients and follow up in both phases to study motality in patients with and without diabetes. While the topic of comorbid diabetes and schizophrenia are highly relevant, and the study design of a matched cohort are in general appropriate, data available are simply too sparse to study mortality in this subgroup. Models are overparamaterized and interpretation of results does not correctly reflect the data presented. Authors should consider to keep the results on the two-way interactions (diabetes – phase), while probably maintain schizophrenia subgroups in the descriptive part of the study, acknowledging that numbers are still too few for statistical inference on these subgroups. I have listed my specific concerns and suggestions below, mainly highlighting the potential implications of estimating mortality rates in the small subgroups.
Major concerns
1. The HRs for the increased hazard rates in the diabetes + schizophrenia group in the two phases of 7.10 (FDS1) and 17.83 (FDS2) (also highlighted in the abstract) are difficult to interpret and compare. This also applies to to the other HRs in Table 1. Why was the ’no diabetes, no schizophrenia’ group in FDS2 used as reference for both phases? The two phases clearly have different length of follow-up, which would likely result in different mortality rates which could likely bias results, as mortality rates may not be constant over time sice a diagnosis of diabetes or schizophrenia. To investigate temporal trends in mortality rates, the same length of follow-up should be applied in the different groups. It is also unclear to me why HRs (point estimates) in FSD2 changed from ~3 to ~18 after adjustment for age in and this age-matched cohort. Are crude mortality rates and rate ratios relevant to present and how should they be interpreted when comparing FDS1 to FDS2?
2. Adjustment for CCI in three levels and sex makes sense for the diabetes groups but not when further stratifying on schizophrenia.
3. Events are too few to present all eight subgroups in a KM plot. A plot of cumulative incidences (absolute risks) divided into subgroups of diabetes and phase would be more meaningful.
4. The results and discussion sections make conclusions based on the point estimates. Although the authors acknowledge the broad confidence intervals in the same sentence, these conclusions and interpretations of estimates should be avoided. Some exaples are
a. Results, p. 7, l. 1-2: ”…showed a three-fold increased risk of death…not reach statistical significance (3.41 (0.59,19.6))”. This result should not be interpreted as a three-fold increased risk.
b. Discussion, first sentense: ”The present study shows that there have been substantial temporal changes in mortality in the 15 years between FDS phases…”. Which results was this based on?
c. Discussion, last paragraph: I do not agree with the authors that ”…FDS data…allowed the assessment of temporal trends…. by type 2 diabetes and schizophrenia status.” and based on the data presented, I am not conviced that ”The mortality gap… increased…”
Minor comments
1. Please be consistent with the use of terms for diabetes and type 2 diabetes in text and tables.
2. How many individuals were censored due to developing schizophrenia during follow-up?
3. Please explain how proportionality of hazards was checked and whether assumptions were fulfilled or not.
1. Why was FSD1 chosen as reference in the statistical analyses (table 2), while FSD2 was used as reference in table 1?
Author Response
Reviewer #1
This study aims at evaluating the temporal changes in mortality in different groups defined by type 2 diabetes and schizophrenia. The topic is very relevant and manuscript is overall well written. A strength is the large dataset with patients recruited in different time periods including sufficient numbers of diabetes patients and follow up in both phases to study motality in patients with and without diabetes. While the topic of comorbid diabetes and schizophrenia are highly relevant, and the study design of a matched cohort are in general appropriate, data available are simply too sparse to study mortality in this subgroup. Models are overparamaterized and interpretation of results does not correctly reflect the data presented. Authors should consider to keep the results on the two-way interactions (diabetes – phase), while probably maintain schizophrenia subgroups in the descriptive part of the study, acknowledging that numbers are still too few for statistical inference on these subgroups. I have listed my specific concerns and suggestions below, mainly highlighting the potential implications of estimating mortality rates in the small subgroups.
Response: Thank you for your constructive insights. Although numbers are small in the schizophrenia sub-groups, we believe the observed trends are sufficiently alarming that they need to be highlighted, albeit interpreted with caution as we hope we have now done.
Major concerns
- The HRs for the increased hazard rates in the diabetes + schizophrenia group in the two phases of 7.10 (FDS1) and 17.83 (FDS2) (also highlighted in the abstract) are difficult to interpret and compare. This also applies to the other HRs in Table 1.
Response: We have revised Table 1 and presented the mortality data separately for each Fremantle Diabetes Study (FDS) Phase as suggested. To make the data more interpretable for the reader, we have used time as the time line rather than age and thus adjusted for age, then additionally for sex and the Charlson Comorbidity Index (CCI) – the version of the CCI we used was an early version without mental health conditions included. For completeness, we also then adjusted for the time-varying interactions of these covariates with ln(time). We have also revised and simplified the Abstract and main text.
Why was the ’no diabetes, no schizophrenia’ group in FDS2 used as reference for both phases? The two phases clearly have different length of follow-up, which would likely result in different mortality rates which could likely bias results, as mortality rates may not be constant over time since a diagnosis of diabetes or schizophrenia. To investigate temporal trends in mortality rates, the same length of follow-up should be applied in the different groups.
Response: As suggested, we have presented results by FDS Phase separately. We are aware that the different lengths of follow-up may bias results and have added that to the limitations section. However, we believe our observations are too serious to wait for longer follow-up in FDS2 before publishing them. However, we have tempered our conclusions as advised. The observed reduction in mortality between FDS1 and FDS2 (36-46%) for all but the schizophrenia subgroup may be attributable to the extended follow-up in FDS1 and the higher mortality as the cohort aged and/or improvements in cardiovascular risk reduction over time reducing mortality. Nevertheless, in FDS2 the combined schizophrenia/type 2 diabetes subgroup remains at an unadjusted 3-fold increased risk of death compared to neither condition (as in FDS1) and when their much younger age is taken into account this increased risk is magnified to a 27-fold increased risk compared to a 6-fold increased risk in FDS1.
- It is also unclear to me why HRs (point estimates) in FSD2 changed from ~3 to ~18 after adjustment for age in and this age-matched cohort. Are crude mortality rates and rate ratios relevant to present and how should they be interpreted when comparing FDS1 to FDS2?
Response: Age (and sex and postcode) matching was made in both FDS1 and FDS2 for the cohorts with diabetes to people without diabetes living in the study area at the time of recruitment of the participant with diabetes to the respective FDS Phase. The FDS1 cohort was significantly younger than the FDS2 cohort; people with schizophrenia were significantly younger than those without schizophrenia. Since age is a major driver of mortality, adjustment for age had a significant effect on the HRs. We have now presented the effect of adjustment for age more clearly by using time as the time line (before we used age as the timeline) and adjusting for age then adding sex and CCI then the time-varying interactions. The HRs for the type 2 diabetes/schizophrenia subgroup in FDS2 using time as the time line and adjusting for age are now 27 versus 3 unadjusted.
- Adjustment for CCI in three levels and sex makes sense for the diabetes groups but not when further stratifying on schizophrenia.
Response: We used the original 17-item CCI derivation (as referenced) which does not include schizophrenia. We have added “…This early version of the CCI did not include mental health conditions” to the Methods Section 2.3.
- Events are too few to present all eight subgroups in a KM plot. A plot of cumulative incidences (absolute risks) divided into subgroups of diabetes and phase would be more meaningful.
Response: We have replaced the KM plot with 4 plots of cumulative mortality by schizophrenia status divided into subgroups of diabetes and phase as suggested.
- The results and discussion sections make conclusions based on the point estimates. Although the authors acknowledge the broad confidence intervals in the same sentence, these conclusions and interpretations of estimates should be avoided. Some examples are
- Results, p. 7, l. 1-2: ”…showed a three-fold increased risk of death…not reach statistical significance (3.41 (0.59,19.6))”. This result should not be interpreted as a three-fold increased risk.
Response: We have modified the text in line with this comment.
- Discussion, first sentense: ”The present study shows that there have been substantial temporal changes in mortality in the 15 years between FDS phases…”. Which results was this based on?
Response: Even though observed absolute mortality rates reduced by approximately 40% in all subgroups except the schizophrenia only subgroup, in FDS2 (as in FDS1) the unadjusted relative risk of death remained 50% increased for those with type 2 diabetes and three-fold increased for those with both conditions compared with the subgroup with neither condition. After age-adjustment, the relative risk was magnified 6-fold in the FDS1 and 27-fold in the FDS2 combined type 2 diabetes/schizophrenia subgroup, suggesting that the mortality gap is at best the same for this vulnerable subgroup in FDS2 as FDS1, and possibly worse. One possible reason is the greater use of obesogenic second generation antipsychotic medication in young people with schizophrenia in the era of FDS2 versus FDS1, accelerating their metabolic dysfunction leading to the development of type 2 diabetes at younger ages. This is now included in the Discussion.
- Discussion, last paragraph: I do not agree with the authors that ”…FDS data…allowed the assessment of temporal trends…. by type 2 diabetes and schizophrenia status.” and based on the data presented, I am not conviced that ”The mortality gap… increased…”
Response: We have reworded the last paragraph with this comment in mind.
Minor comments
- Please be consistent with the use of terms for diabetes and type 2 diabetes in text and tables.
Response: We have edited the manuscript and tables as suggested.
- How many individuals were censored due to developing schizophrenia during follow-up?
Response: Two in total, one from the FDS1 type 2 diabetes cohort, one from the FDS2 no diabetes cohort. They were both aged in their early 70s at first mention of schizophrenia in their medical records, so the diagnoses may be late or erroneous, but we have erred on the side of caution and censored their follow-up at the time of first documentation of schizophrenia.
- Please explain how proportionality of hazards was checked and whether assumptions were fulfilled or not.
Response: We have added the following to the statistical methods section:
“The proportional hazards assumption was checked using time-varying covariates. When this assumption was violated, a time-varying interaction of each covariate with ln(time) was added to the model”.
Since the covariates largely violated the proportional hazards assumption, we have additionally adjusted for their time-varying interactions with ln(time) in Table 1. This made little difference to the results.
- Why was FSD1 chosen as reference in the statistical analyses (table 2), while FSD2 was used as reference in table 1?
Response: We have removed Table 2.
Reviewer 2 Report
The present study explored the temporal trends in mortality associated with comorbid type 2 diabetes and schizophrenia, by using the Fremantle Diabetes Studies 1 and 2 (FDS1 & FDS2). They found that combination of type 2 diabetes and schizophrenia was associated with a substantial and possibly increasing mortality gap. I have several comments.
1. Schizophrenia was uncommon, affecting 67/13,995 (0.48%) people overall and 17/2,800 (0.61%) of those with type 2 diabetes. The sample size of schizophrenia patients remained too much lower. How about any other evidences or literatures support the association between schizophrenia and type 2 diabetes in mortality?
2. Considering the schizophrenia and type 2 diabetes or other metabolism disorders, which is the cause or consequence, leading to the mortality?
3. How are the confounding effects antipsychotic medicines on the mortality in the schizophrenia patients comorbid type 2 diabetes?
4. The authors just used the semi-structured interviews for schizophrenia patients according to ICD-9 or ICD-10. That’s not enough for the diagnoses.
Author Response
Reviewer #2
The present study explored the temporal trends in mortality associated with comorbid type 2 diabetes and schizophrenia, by using the Fremantle Diabetes Studies 1 and 2 (FDS1 & FDS2). They found that combination of type 2 diabetes and schizophrenia was associated with a substantial and possibly increasing mortality gap. I have several comments.
- Schizophrenia was uncommon, affecting 67/13,995 (0.48%) people overall and 17/2,800 (0.61%) of those with type 2 diabetes. The sample size of schizophrenia patients remained too much lower. How about any other evidences or literatures support the association between schizophrenia and type 2 diabetes in mortality?
Response: We agree that the numbers with schizophrenia were small and have acknowledged this as a limitation in the Discussion. We have cited other relevant papers in the Introduction and Discussion.
- Considering the schizophrenia and type 2 diabetes or other metabolism disorders, which is the cause or consequence, leading to the mortality?
and
- How are the confounding effects antipsychotic medicines on the mortality in the schizophrenia patients comorbid type 2 diabetes?
Response: We have expanded the relevant parts of the Discussion in view of these comments. Specifically, overall improvements in mortality were seen in the FDS2 compared with the FDS1, but the mortality gap due to schizophrenia was evident in both studies and the combination of type 2 diabetes and schizophrenia was associated with a substantially increased risk of death. This latter observation is likely multifactorial, perhaps reflecting less effective cardiometabolic management in those with schizophrenia compounded by the adverse effects of antipsychotic therapies
- The authors just used the semi-structured interviews for schizophrenia patients according to ICD-9 or ICD-10. That’s not enough for the diagnoses.
Response: Please refer to the Methods Section 2.3 in which we describe in detail the ascertainment of prevalent and incident schizophrenia, which was not through semi-structured interview but diagnostic codes in two administrative data collections:
“Linkage through the WADLS provided validated data on prior history and incident events. In WA, the Hospital Morbidity Data Collection (HMDC) contains information regarding all public/private hospitalizations since 1970, the Mental Health Information System (MHIS) started as a register of psychiatric inpatients in 1966, was expanded to include hospitals and community mental health services in the 1970s and, since 1980, has covered all outpatient, community-based and hospital-based mental health services…. Relevant International Classification of Disease (ICD)-9-CM (295) and ICD-10-AM diagnosis codes (F20) were used to identify prevalent and incident schizophrenia in the HMDC and MHIS”.
In the Discussion we acknowledge this as a limitation but reference the validation of the coding through semi-structured interview.

Reviewer 3 Report
i appreciate your work
Author Response
No response required
Reviewer 4 Report
The study examined the combined impact of schizophrenia and diabetes on mortality risk and investigated whether the mortality gap had improved using a representative Australian urban sample population. After reviewing the manuscript (which has probably gone through a previous round of revision), the authors have appropriately presented their study. It was well-written and addresses an important topic in the interplay of diabetes and psychiatric conditions. The relatively small number of participants with schizophrenia and diabetes is not limited to the current paper but could also be due to the low prevalence of schizophrenia in the population (This could be added to the manuscript, with a reference to a recent prevalence report). The implication is that a very large study is needed.
Author Response
Reviewer #4
The study examined the combined impact of schizophrenia and diabetes on mortality risk and investigated whether the mortality gap had improved using a representative Australian urban sample population. After reviewing the manuscript (which has probably gone through a previous round of revision), the authors have appropriately presented their study. It was well-written and addresses an important topic in the interplay of diabetes and psychiatric conditions. The relatively small number of participants with schizophrenia and diabetes is not limited to the current paper but could also be due to the low prevalence of schizophrenia in the population (This could be added to the manuscript, with a reference to a recent prevalence report). The implication is that a very large study is needed.
Response: In response to the Reviewer’s suggestion, we have included, in the limitations paragraph towards the end of the Discussion, some general population estimates of schizophrenia prevalence which align with our rate in the participants without diabetes.
Round 2
Reviewer 1 Report
Thank you for the revised manuscript. I acknowledge that the authors have revised the manuscript following some of my advise, e.g. revised table 1, avoiding interactions and looking into model assumptions. However, the main problem still remain; the number of deaths are simply to few in the comorbid schizophrenia and diabetes groups to make robust statistical inference and make sound conclusions based on estimated mortality rate ratios withe 95% confidence intervals. The extremely high point estimate (after adjustment for age) of 27 and with a 95% confidence interval ranging from 10 to 73, should not be reported as a 27-fold increased risk. Together with the non-proportionality of hazards it reflects that this high estimate is driven by very few cases dying at young ages in the comorbid group. I do understand the rationale for including interactions with time to account for the violated model assumptions, but the number of deaths in some exposure groups are too low to include that many parameters in the model. Furthermore, a recently published observational study in the Danish population of nearly 6 mill. People found a mortality rate ratio of 4.01 (95% CI 3.89-4.13) comparing people with schizophrenia and diabetes to those with neither of the two disorders.(1) As I stated in my first review report, ”Authors should consider to keep the results on the two-way interactions (diabetes – phase), while probably maintain schizophrenia subgroups in the descriptive part of the study, acknowledging that numbers are still too few for statistical inference on these subgroups. I have listed my specific concerns and suggestions below, mainly highlighting the potential implications of estimating mortality rates in the small subgroups”. In other words, I do think that the topic is highly relevant and data valuable, but in groups with sparse data I would only include the descriptive part, i.e. numbers and not estimating (adjusted) rate ratios based on a Cox proportional hazards model.
1. Momen NC, Plana-Ripoll O, Agerbo E, Christensen MK, Iburg KM, Laursen TM, et al. (2022): Mortality Associated With Mental Disorders and Comorbid General Medical Conditions. JAMA Psychiatry. 79:444-453.
Author Response
Thank you again for your insightful comments and the recent reference.
However, with regards to the Momen et al. reference, the cohorts from this reference and the Fremantle Diabetes Study (FDS) Phase I and II are very different and thus not comparable:
- The frequency of schizophrenia and related disorders in Momen et al. seems high at 1.5% compared with the 12-month treated prevalence of schizophrenia/schizoaffective disorder in Australian adults aged 18-64 years (0.3%) (VA Morgan et al. Australian & New Zealand Journal of Psychiatry, 2012; 46(8): 735-752). The prevalence of schizophrenia among the 18-64 year olds (both those with type 2 diabetes and their matched non-diabetic counterparts) in our community-based cohort (with a different method of ascertainment and likely different age profile to the survey) was 0.5% in FDS1 (recruited 1993-6) and 0.7% in FDS2 (recruited 2008-11).
- The ascertainment of diabetes for the Momen et al. reference’s cohort from hospital admissions and prescription data is sub-optimal because it excludes those who have not been hospitalised for/with diabetes and a significant proportion of people living with type 2 diabetes who do not require medication to manage their hyperglycaemia.
- There is no stratification by type of diabetes. This is very important since type 2 diabetes is mostly diagnosed in middle-late adulthood, but the use of obesogenic second generation antipsychotic medication accelerates the development of type 2 diabetes at much younger ages in those living with schizophrenia.
- The age profiles of the Momen et al. reference’s cohort and our two cohorts were quite different being based on different methodologies; for example, the reference’s cohort median [IQR] age was 32.0 [7.3-52.9] years at the start compared with 65.0 [57.3-71.9] years for FDS1 and 65.8 [57.7-74.2] years for FDS2; the reference’s cohort median [IQR] age at census was 48.9 [42.5-68.8] years compared with 79.5 [72.6-85.2] years for FDS1 and with 72.7 [64.4-80.7] years for FDS2.
- Under the limitations for the Momen et al. reference, the authors state that, “Furthermore, it is likely that our findings have limited generalizability outside of Denmark: patterns of comorbidity and mortality vary between countries, particularly among those with different health care and socioeconomic structures.”
Having said that, the Momen et al. reference is a substantial piece of work and we share the conclusion that much more needs to be done to support people living with psychotic illness (specifically schizophrenia in our paper) and manage comorbidities (specifically type 2 diabetes in our paper) optimally. Where we differ is in our ability to compare mortality outcome in people with both type 2 diabetes and schizophrenia compared with neither condition in two community-based cohorts recruited 15 years apart. Despite small numbers, as we have acknowledged in our paper, we believe there is a sufficiently strong signal to warn that the mortality gap in those living with both conditions compared with neither condition has at best stayed the same, but at worst widened, due to the increasing use of obesogenic second generation antipsychotic medication in people living with schizophrenia.
We beg to differ with respect to presenting the Cox model adjusted for age, since people are generally diagnosed with schizophrenia in their late teens/early twenties whilst conventionally people were diagnosed with type 2 diabetes in middle-late age. But the introduction of obesogenic second generation antipsychotic medication in people living with schizophrenia has accelerated the onset of type 2 diabetes in young people with schizophrenia. This is especially evident between the two phases of the Fremantle Diabetes Study when age is adjusted for. Even unadjusted, the 95% confidence intervals do not span unity and thus there is no type II error; we have sufficient power despite the small numbers because the effect size is so large.
In deference to the Reviewers’ concerns, however, we have added a comment in the Discussion that our work needs to be replicated in much larger studies.
Reviewer 2 Report
All my comments have been well addressed. This is an interesting study, and I recommend to publish it.
Author Response
No response required